# Three-dimensional, printed water-filtration system for economical, on-site arsenic removal

**Kihoon Kim**[1☉], **Monica Cahyaning Ratri**[1,2☉], **Giho Choe**[1], **Myeongyun Nam**[1], **Daehyoung Cho**[1], **Kwanwoo Shin**[1]*

**1** Department of Chemistry and Institute of Biological Interfaces, Sogang University, Seoul, Republic of Korea, **2** Department of Chemistry Education, Sanata Dharma University, Yogyakarta, Republic of Indonesia

☉ These authors contributed equally to this work.
* kwshin@sogang.ac.kr

**Data Availability Statement:** All relevant data are within the manuscript and its Supporting Information files.

**Funding:** This research gets support from the Basic Science Research Program

## Abstract

The threat of arsenic contamination to public health, particularly in developing countries, has become a serious problem. Millions of people in their daily lives are still highly dependent on groundwater containing high levels of arsenic, which causes excessive exposure to this toxic element, due to the high cost and lack of water-treatment infrastructures. Therefore, a technique for large-scale treatment of water in rural areas to remove arsenic is needed and should be low-cost, be easily customized, and not rely on electrical power. In this study, in an effort to fulfill those requirements, we introduce a three-dimensional (3D), printed water-filtration system for arsenic removal. Three-dimensional printing can provide a compact, customized filtration system that can fulfill the above-mentioned requirements and that can be made from plastic materials, which are abundant. Armed with the versatility of 3D printing, we were able to design the internal surface areas of filters, after which we modified the surfaces of the 3D, printed filters by using iron (III) oxide as an adsorbent for arsenite. We investigated the effects of the controlled surface area on the flow rate and the deposition of the adsorbent, which are directly related to the adsorption of arsenic. We conducted isotherm studies to quantify the adsorption of arsenic on our 3D, printed filtration system.

## Introduction

Arsenic is a natural element on Earth and is well known to be a toxic material for humans and other living organisms and to cause serious health problems in many areas of the world. Arsenic exists in various oxidation states: As(III) called arsenite and As(V) called arsenate; these inorganic forms of arsenic are very harmful to living organisms. Arsenite poses a greater threat to humans than arsenate, and long-term exposure to inorganic arsenic can lead to chronic arsenic poisoning and cause human diseases, including cancer, of the lungs, bladder, liver, skin, kidneys, and cardiovascular system, as well as gastrointestinal disturbances and neurological disorders [1]. Due to the harmful effects of arsenic on humans, the World Health

(2018R1A6A1A03024940) through the National Research Foundation (NRF) funded by the Ministry of Science and ICT (MSIT), Korea. http://www.nrf.re.kr The funders had no role in study design, data collection and analysis, decision to publish, or preparation of the manuscript.

**Competing interests:** The authors have declared that no competing interests exist.

Organization (WHO) has established as a drinking-water standard a provisional guideline for the concentration of arsenic in water of 0.01 mg/L [2]. Approximately 137 million people in 70 countries have been affected by arsenic-contaminated drinking water [3]: In China, 14.7 million people are exposed to arsenic contamination of over 0.01 mg/L in their drinking water [4], and 1.1 million people in Kandal Province, Cambodia [5] and about 60–100 million people in India and Bangladesh have been estimated to be currently at risk as a result of drinking arsenic-contaminated water [6]. The threat to public health in developing countries is more serious than it is in developed countries because, in developing countries, household water supplies mainly come from natural groundwater sources, which contain high levels of arsenic [7]; thus, this widespread health risk in developing countries is a serious problem of immediate concern.

Arsenic removal by adsorption is potentially applicable for small-scale water treatment due to its simplicity, cost-effectiveness, and ease of use [8]. Adsorptive filtration is one of the adsorption processes for removing inorganic arsenic species from water and is often combined with other technologies, such as coagulation flocculation, oxidation, and microbial sanitation. In adsorptive filtration, contaminated water flows through an adsorptive medium, where toxic species are immobilized and removed [9]. Adsorptive filtration for arsenic removal is a convenient and simple post-treatment technique, making it appropriate for daily use; moreover, an adsorptive filtration system is reusable and offers many advantages, particularly in the time and the cost of operation [10, 11]. Among various arsenic absorbents, iron oxide is one of the most promising adsorbents because it exhibits selective affinity for arsenic ions and is a very cheap material that is abundant in nature. Therefore, iron oxide may be a very effective alternative for domestic water treatment when used as an adsorbent in a filtration system [12, 13]. Much research has been done on iron-oxide-modified media, including iron-oxide-coated zeolite [14], cement [15], carbon nanotubes (CNTs) [16], activated carbon [17], biomasses [18], activated alumina [19], and polymers, for arsenic removal [20]. Iron-oxide-coated sand, which is composed of sand grains as cores and iron oxide as an outer layer for immobilization and removal of arsenic, is a widely known adsorptive medium for arsenic removal [21]. While the above approaches efficiently reduce the arsenic concentration in water, the application of such methods in rural areas is still limited due to the cost of materials, the lack of expertise, the large-scale intricacy of the removal system, and the lack of flow control without electricity.

A technology that can potentially be used to fulfill the need for a process that can filter arsenic from water at low cost and be controlled without electricity is 3D printing. Three-dimensional printing enables small quantities of robust, innovative, and customized goods to be rapidly produced at relatively low costs and customized and complex objects to be produced on micrometer-to-millimeter scales. This technology can also be used without the need for the processes used in traditional manufacturing systems, such as milling or molding [22]. Three dimensional printing is used in fields such as biomedical engineering [23], electronics [24], microfluidics [25], synthetic biology [26] and soft robotics [27]. In developing countries, 3D printing is applied to solve local problems, e.g., the 3D printing of architectures; clothing; basic medical supplies such as prosthetic limbs in Uganda and South Sudan [28, 29]; umbilical clamps, finger splints, and casts in Haiti [30]; and a special 3D printed vein finder in Kenya [28]. Three-dimensional, printed devices, which can be designed on demand, are being used for environmental safety; moreover, water filters using 3D printed clay have been developed [31] and specific filters with good adsorption capacity and effective separation have been designed and fabricated [11].

A 3D, printed device involving chemical reactions is called '3D printed reactionware'. Such 3D, printed reactionware has potential to be used as a cheap, automated, and reconfigurable chemical-reaction platform [32]. Plastic, which is abundant and cheap, is one of the materials

that can be used as reactionware in 3D printing. As one of the most popular printing methods, fused deposition modeling (FDM) involves melting thermoplastics through a heated nozzle and depositing them layer-wise. Hence, customized fabrication of desired products at low cost can overcome deficiencies in the supplies of essential goods in developing countries.

In the research, we successfully manufactured a 3D, printed water-filtration system for the removal of arsenite. Owing to the versatility of 3D printing, the filter's features, such as size and shape, could be easily changed to meet the need at any particular contaminated site. The filtration system was composed of a 3D, printed polylactic acid (PLA) filter coated with iron (III) oxide for arsenite removal and square channels with controlled widths for management of the water flow. Iron (III) oxide was directly deposited onto the 3D printed filter, and as the contaminated water flowed through the filter, arsenite was then adsorbed onto the iron oxide on the filter's surface. The width of the filter's channels could be altered, resulting in filters with different surface areas and volumes. Changes in the surface area and the volume enabled control of the amount of iron (III) oxide that was coated onto the filter, consequently leading to differences in the efficiencies of water purification. The water retention time was different for filters with different channel surface areas and different volumes. For slower water flow, 3D, printed channels could be made to have smaller sizes and denser packing, and vice versa for faster flow. We investigated the effects of the amount of iron (III) oxide on arsenite removal and of the channel width on the flow rate and the removal efficiency. This novel approach should enable the fabrication of small-scale, low-cost, customized water-filtration systems that can used without the need for electricity.

## Materials and methods

### Materials

Sodium (meta) arsenite ($NaAsO_2$), iron (III) chloride ($FeCl_3$), leucomalachite green ($H_5CH$ $[C_6H_4N(CH_3)_2]_2$) and hydrochloric acid (HCl) were all purchased from Sigma-Aldrich (USA). The PLA 3D filament was purchased from eSUN (China), sodium acetate ($CH_3COONa$) from Fluka (USA), acetic acid ($CH_3COOH$) from Jinchemical Co., Ltd (Korea), sodium hydroxide (NaOH) from Samchun Pure Chemical Co., Ltd (Korea), and potassium iodate ($KIO_3$) from Junsei Chemical Co., Ltd (Japan).

### Methods

**Three-dimensional modeling and printing.** Models of the filters were developed using AutoCAD 2016 (Autodesk, USA). While the exteriors of the filters were identically designed to be 40-mm high (with a 20-mm long guard) and to have a 30-mm outer diameter, the channel widths within a cylinder with a 27-mm inner diameter were varied from 0.8 to 4.0 mm (Fig 1A). The CAD models were exported from AutoCAD in the form of stereolithography (STL) data. STL data were then imported to Simplify3D (Simplify3D LLC, USA), a software that converted the STL data into instructions for the 3D printer to print according to the selected settings. The 3D printer that was used was a DeltaBot unit (Lab 311, Korea). The nozzle was heated to melt and deposit the PLA filament through the calculated pathway.

**Surface treatment of three-dimensionally printed PLA filters (hydrolysis and iron-oxide coating).** Surface treatment was carried out using hydrolysis and iron-oxide coating, as shown in Fig 1B. Hydrolysis was done using a 10-M HCl solution, which was poured through the filter channel to hydrolyze the polyester group in PLA. The iron-oxide-coating process was begun by using $FeCl_3$ and NaOH in an aqueous solution to precipitate iron (III) hydroxide ($Fe(OH)_3$). While 300 mL of 0.025-M sodium hydroxide was being rapidly stirred at 700 rpm, 400 mL of 0.025-M $FeCl_3$ was slowly added and stirred for another 1 min. Fe

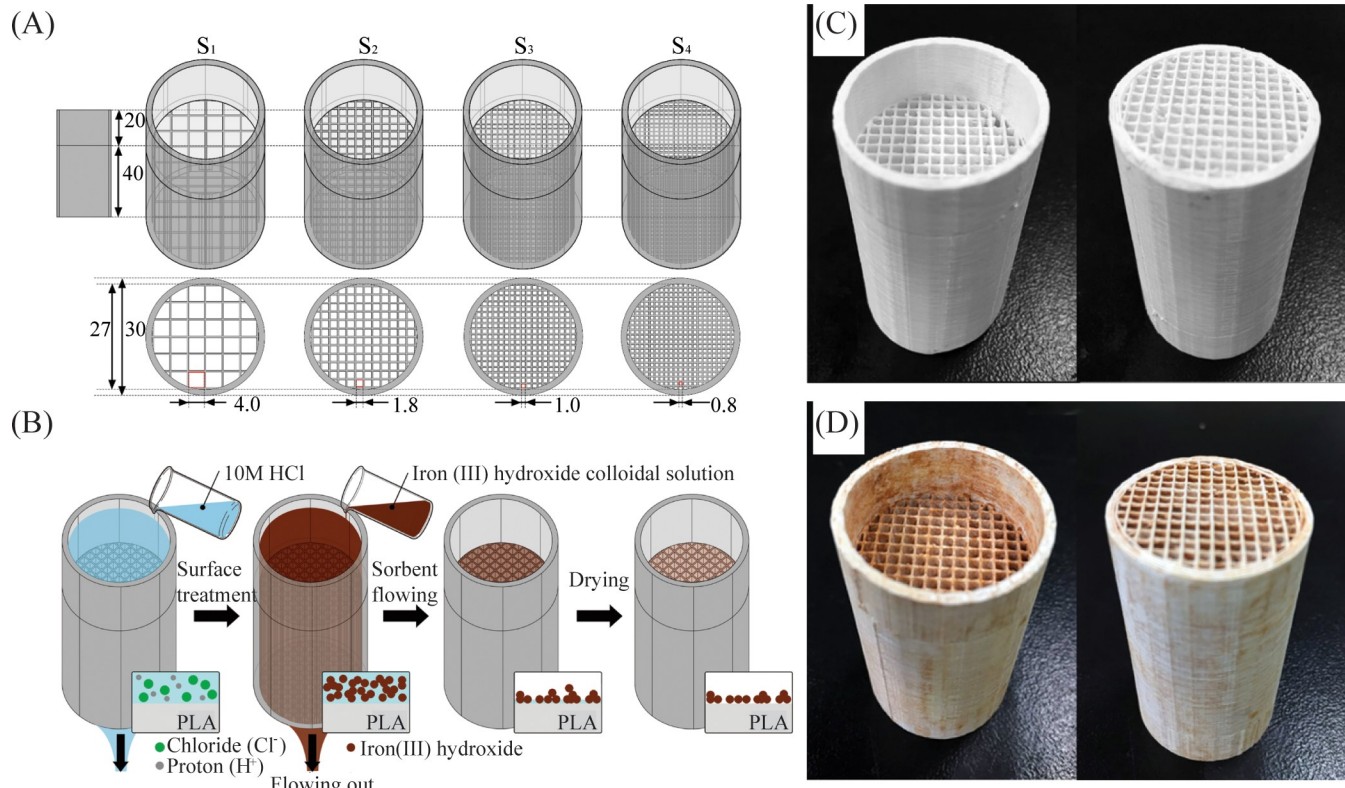

**Fig 1. Three-dimension filter design and illustration of sample preparation.** (A) Computer-aided designs of the filters. Four differently sized filters were designed, and the channel widths of the unit squares were 0.8, 1.0, 1.8 and 4.0 mm. All the numbers in the figure are in millimeters. (B) Illustration of the sample preparation process on the surface of a 3D, printed PLA filter: acid hydrolysis, iron-oxide loading and drying. (C, D) Pictures of the top (left) and the bottom (right) of a 3D, printed filter (C) before and (D) after the adsorptions of iron (III) oxide.

$(OH)_3$ particles are highly insoluble in water, so in order to increase the deposition of $Fe(OH)_3$ onto the filters, we maximized the insolubility by optimizing the pH [33].The pH was adjusted to 8.0 by carefully adding 0.1-M HCl and NaOH into the solution while stirring. In a few minutes, $Fe(OH)_3$ particles had precipitated to the bottom of the beaker. The supernatant was removed to increase the concentration of $Fe(OH)_3$ until 300 mL of solution remained. The 3D, printed PLA filters that had been treated with HCl previously were then dipped into and lifted out of the solution five times (S1 File). The filters were dried in an oven at 60°C for 12 hours. After the drying process, 1 L of tap water was passed through each filter to eliminate unbound $Fe(OH)_3$ particles inside. The soaking, lifting, drying, and washing processes were repeated once more. Images of the filters before and after the sorbent loading are presented in Fig 1C and 1D. The adsorption of iron (III) oxide comes in a rust-like form on the surface of the filter, which is brown in color because the original color of the iron (III) oxide solution is brown; the change in the filter's color confirms that iron (III) oxide has been absorbed onto the PLA filter.

The amount of $Fe(OH)_3$ loaded on the filter was measured by weighing each filter before and after the loading process. In order to confirm the chemical compositions of $Fe(OH)_3$ that had been coated on the filters, we used scanning electron microscopy—energy dispersive X-ray spectroscopy (SEM-EDX; JSM-7100F, JEOL Ltd.) and X-ray photoelectron spectroscopy (XPS; K-alpha, Thermo Scientific Inc.) to examine the chemical components of the filters before and after the adsorptions of the iron (III) oxide. An experiment recycling the 3D printing filter used in the experiment for the adsorption of As was carried out in the following

manner: First, the As (III) and the iron-oxide (III) complexes were removed by filtration with 20% $H_2SO_4$ solution. As in the above, the process was repeated in the same manner as the iron oxide was adsorbed.

**Batch kinetic experiments.** The reaction rate was calculated by measuring the decrease in arsenic concentration with increasing contact time. The bottom of each filter was blocked with paraffin film, and each filter was filled with 15 mL of the standard arsenite solution (20 mg of As per L of deionized water, pH 8.0) at 25ºC. An UV-VIS spectrophotometer (V-660, JASCO) with leucomalachite green dye and potassium iodate was used to measure the arsenite concentration for high concentrations $\geq 0.2$ mg/L [34]. An inductively coupled plasma mass spectrometer (NexION300, PerkinElmer) was used to measure the arsenite concentration for low concentrations < 0.2 mg/L.

**Isotherm experiments.** Water containing arsenite was kept stationary in each $Fe(OH)_3$-loaded filter for 3 hours at 25ºC, the time reported to be sufficient for reaching chemical equilibrium. The extent of arsenic removal was then measured using samples with different arsenite concentrations. The optimal pH for efficient arsenite removal was reported to be 8.0, which is representative of groundwater [35, 36], so the pH was adjusted to 8.0 for all experiments. The cavities of the iron (III)-oxide-loaded filters were sealed with paraffin film to prevent the evaporation of the arsenic solution. Equilibrium between the arsenite and the iron (III) oxide was reached after 2 hours [21]. After the solution had been kept in the filters for 3 hours, it was extracted, and the amount of remaining arsenic was measured using either the UV-VIS spectrophotometer or the inductively coupled plasma mass spectrometer (NexION300, PerkinElmer).

**Column experiments.** The flow rate was calculated by measuring the time during which a certain amount of water flowed through filters with different channel widths. The filter was connected to the neck of a 500-mL volumetric flask containing tap water, and water was flowed through the filter. The connection between the filter and the flask was carefully sealed to prevent the inflow of air. After the flask had been filled with water, it was turned upside down, and the time required for the water to penetrate the filter was measured. For consistent measurements, we flowed 10 mL of arsenic solution through filters with various channel widths. A volumetric flask containing 500 mL of standard arsenite solution (20 mg of As per L of deionized water, pH 8.0) was connected to the $Fe(OH)_3$-loaded filter and was again sealed with paraffin film. A 3D, printed rate controller was attached to each end of the filter (S2 Fig). The flask connected to the filter was then turned upside down and was kept stationary until 10 mL of water had flowed out at 25ºC. Then, the concentration of arsenite in that water was measured using either the UV-VIS spectrophotometer or the inductively coupled plasma mass spectrometer.

## Results and discussion

### Three-dimensional modeling and printing

A difference in the filter's channel width leads to differences in the internal surface area and the total channel volume. The internal surface areas and the total channel volumes of the 3D objects were calculated by using the AutoCAD, MASSPROP and AREA commands. Fig 2A shows the effects of channel width variation on the internal surface area and the total channel volume. As the channel width increases, the internal surface area decreases, but the total channel volume increases. A smaller meshed filter with a higher surface area will offer greater filtering ability but has less water capacity.

### Surface treatment of filters (hydrolysis and iron-oxide coating)

As illustrated in Fig 2B, the fabricated PLA filter was then hydrolyzed using a strong acid. Placing the PLA filter in an acidic environment leads to the dissociation of the ester linkages of

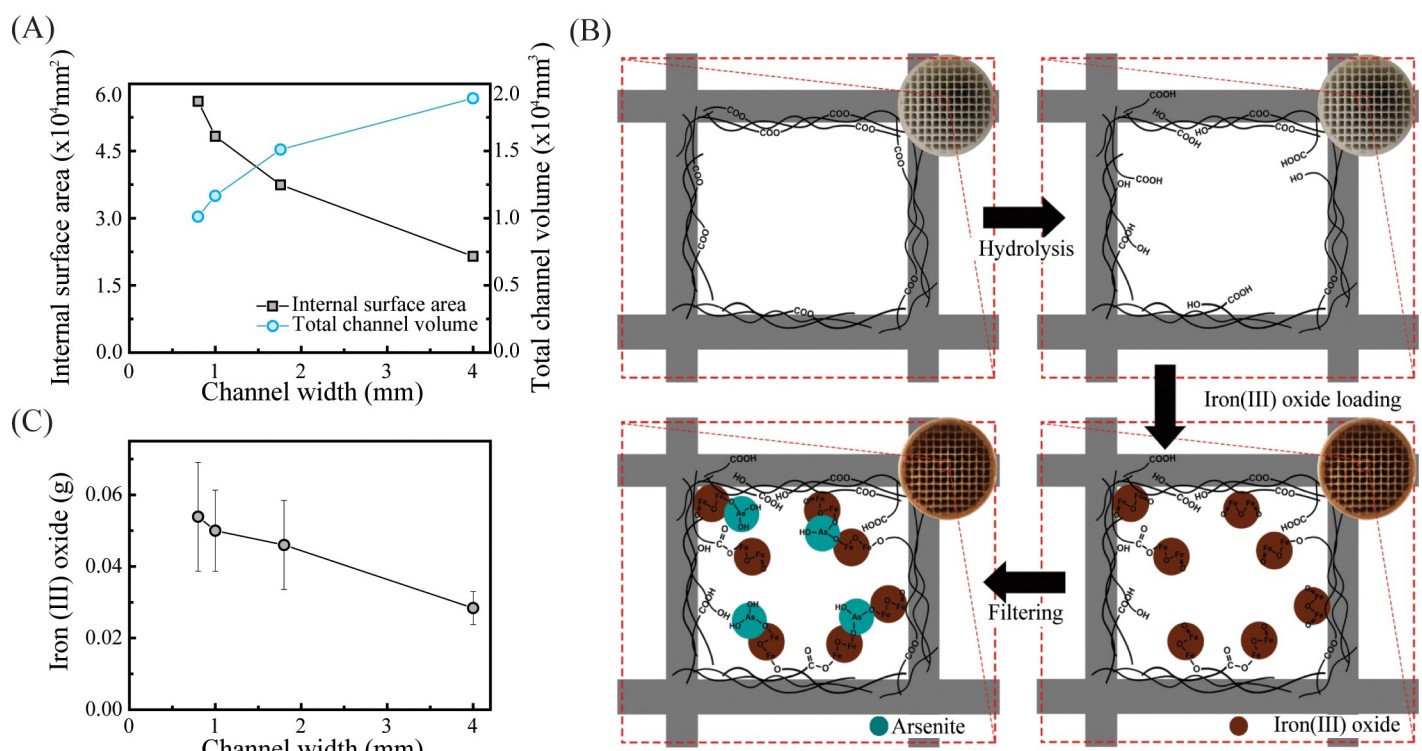

**Fig 2. The iron (III)-oxide coating onto the 3D printed channels.** (A) Calculated internal surface area and total channel volume of the filters as a function of the channel width. The internal surface area represents the total area of the square channels inside the filters (blue circles). The total channel volume is the sum of the empty spaces in the filters that can be occupied by eluent during the filtering process (black squares). (B) Cross-sectional schematics of the sample preparation and filtration processes. Actual photographs are shown in the insets. Upon acidic hydrolysis, carboxylic and hydroxyl groups are formed inside the channels. The hydrolyzed filters are then soaked in an iron (III)-hydroxide solution and dried, leading to iron (III)-oxide deposition onto the surface. The deposited iron (III) oxides are simplified as a formula unit (brown circles). After the sample preparation steps, As (III) solution flows through the filters, and As (III) ions (blue circles) are adsorbed onto the iron (III) oxide. (C) Amount of iron (III)-oxide loading of the filters as a function of the channel width.

PLA [37–40], yielding hydroxyl and carboxylic groups, as the chemical reaction is shown in Fig 3 [41]. Then, the increased hydrophilicity of the channels facilitates water flow and increases the amount of $Fe(OH)_3$ deposition. The $Fe(OH)_3$ was precipitated by a reaction between $FeCl_3$ and $NaOH$, which can be expressed as

$$Fe^{3+}_{(aq)} + 3OH^-_{(aq)} \rightarrow Fe(OH)_{3(s)}. \tag{1}$$

Fig 2B presents schematics of the hydrolysis and the filtration processes, resulting in the selective adsorption of As(III) on the filter's surface. The hydrolyzed filter was soaked in an iron (III)-hydroxide solution and dried, and HCl and water were evaporated during the heating step [42], leading to iron (III)-oxide deposition on the surface of the filter. Formation of

**Fig 3. Dissociation reaction of the ester linkages of PLA.**

those functional groups (hydroxyl and carboxylic acid) and iron (III)-oxide deposition on the surface of the filter were further confirmed by using XPS, and the results are shown in S1 Fig and S4 Table. The hydrolyzed PLA and iron (III)-oxide deposition were clearly confirmed by the O (1s), C (1s) and Fe (3d) spectra in S1(A), S1(B) and S1(C) Fig. An As(III) solution was then flowed through the filter, and As(III) ions were adsorbed onto the iron (III) oxide via inner-sphere complex formation, resulting in the formation of ferric-arsenite complexes. The mechanism for the sorption process is as follows [20]:

$$M-FeOH + H_3AsO_3 \rightarrow M-Fe-H_2AsO_3 + H_2O \tag{2}$$

The channel width also plays a role in increasing the amount of sorbent loading (Fig 2C). As the channel width is decreased, the channel becomes narrower, resulting in a larger internal surface area on which the iron (III) oxide can be immobilized; thus, the amount of iron (III) oxide adsorbed on the channel's surface is increased [43].

In order to confirm the adsorption of arsenic ions to iron oxides, we performed elemental analyses. The SEM-EDX mapping of the Fe and the As elements on the surface of a filter after filtration can be seen in Fig 4A. Iron was evenly distributed over the entire area of the 3D, printed PLA filter, and the adsorption of filtered As was confirmed. Fig 4B and 4C show the filter's surface before and after arsenite filtration, respectively; after filtration, the roughness of the filter's surface was increased, and adsorptive elements were seen upon the filtration. The adsorption of arsenite on the iron-oxide-coated filter was quantitatively analyzed through an EDX analysis. While no arsenic was found before the filtration (Fig 4D), a substantial amount of arsenic of 0.4 weight percent was found on the surface of the filter. The filtration of As (III) was further confirmed by using XPS, and the results are shown in S1(D) Fig.

## Batch kinetic study

A batch experiment was conducted to investigate the kinetics of adsorption and the adsorption capacity for a small amount of solute in the solution. Normally, batch experiments are conducted through agitation at a constant rate and temperature [44]. However, for our 3D,

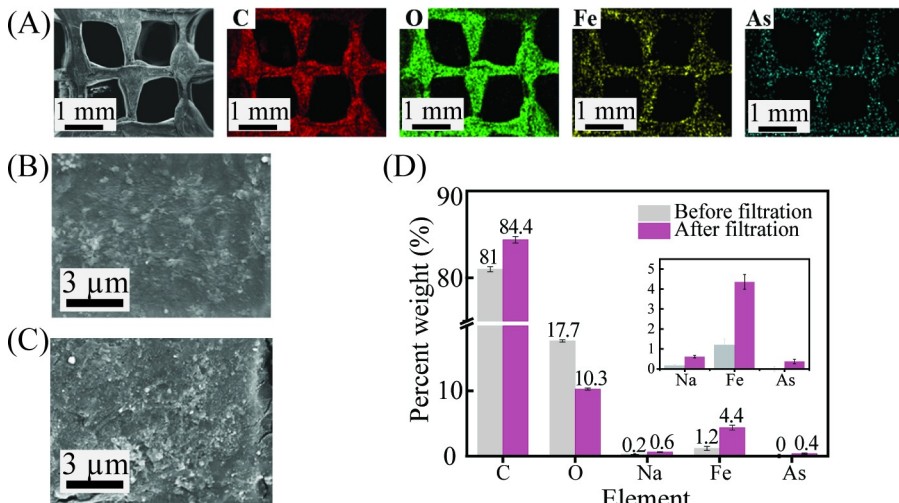

**Fig 4. SEM-EDX images and elemental analysis of the 3D printed filters.** (A) SEM images and EDX elemental mapping (carbon (C), oxygen (O), iron (Fe), and arsenic (As)) of the filter after filtration. SEM images of the surface of the filter (B) before and (C) after As (III) filtration. (D) Quantitative distribution of elements by weight percent before and after As (III) filtration.

printed filtration system, such batch studies cannot be applied because the adsorbent is immobilized on the filter's surface and can be neither separated nor agitated in the solution. Therefore, our batch studies were conducted in a different way; the bottom of the filter was blocked, and the filter was filled with arsenite solution at 25ºC without agitation.

The adsorption of arsenic can be described using kinetic models, and in Fig 5A, the effect of the channel width on the removal efficiency of arsenic is described. The batch study was conducted at room temperature, the pH was adjusted to 8.0, and the initial concentration in deionized water was 20 mg/L. In this study, the pH was adjusted to 8.0 to form a negatively charged state of iron (III) oxide. For a given reaction time, filters with narrower channels showed enhanced removal performances compared to filters with wider channels. Fig 5A shows that the adsorption equilibrium was reached after 2 minutes for the filters with 0.8-mm and 1.0-mm channel widths; a longer time was required to reach equilibrium for filters with wider channels. Initially, the absorption process was very rapid because the As(III) easily adsorbed at the iron (III)-oxide sites in the modified channel, but as time passed, the rate of absorption decreased due to the iron (III)-oxide sites in the modified channel being increasingly occupied by As(III). In order to investigate this phenomenon in a mathematical way, we adopted pseudo-second-order kinetics because of the limiting rate of absorption caused by chemical adsorption, commonly called chemisorption; this absorption phenomenon involves an exchange, or sharing, of an electron or electrons between the sorbate and the sorbent through covalent bonding [45, 46]. The most common form of the pseudo-second-order equation can be expressed in differential form as

$$\frac{dq_t}{dt} = k_2(q_e - q_t)^2, \tag{3}$$

where $q_t$ (mg/g) is the amount of As(III) adsorbed per unit weight of iron (III) oxide at time $t$ (min), $k_2$ is the rate constant (g/mg·min), and $q_e$ is the amount of solute adsorbed at equilibrium (mg/g). After integrating Eq (3), applying the boundary condition $q_{t=0} = 0$, and rearranging terms, we find

$$\frac{t}{q_t} = \frac{1}{k_2 q_e^2} + \frac{t}{q_e}. \tag{4}$$

A plot of $t/q_t$ against $t$ (Fig 5B) yields a linear relationship from which $q_e$ and $k_2$ can be obtained from the slope and the y-intercept of the plot, respectively.

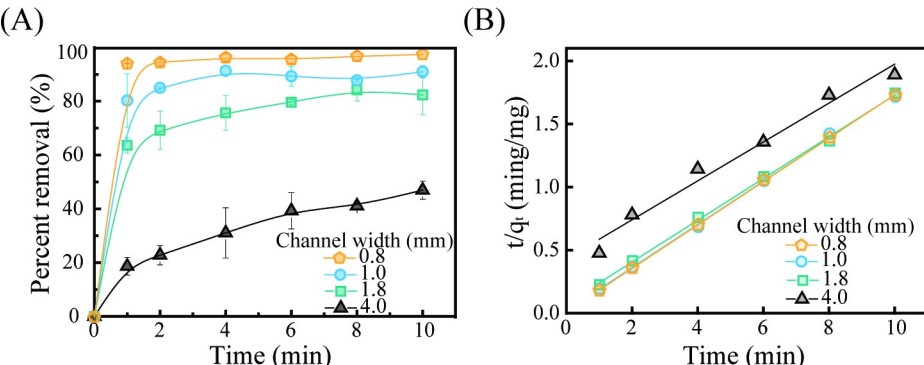

**Fig 5. Batch kinetics for As (III) removal.** (A) The effect of the contact time on As (III) removal. (B) A plot of t/qt versus time (t) for calculating the kinetic parameters. Regression curves are based on pseudo-second-order predictions. The initial concentration of As (III) was 20 mg/L. The pH was adjusted to 8.0 and the batch study was conducted at room temperature.

The data show that filters with narrower channels have higher values of $k_2$ and $q_e$ (S1 Table). This condition indicates that as the channel gets narrower, removal of arsenic is faster and better, with the regression curve showing high significance ($R^2 > 0.97$). The value of $k_2$ for the 0.8-mm channel width is 30 times greater than the value for the 4.0-mm channel width. The difference in the values of $k_2$ originates from the differences in the surface areas, total channel volumes, and amounts of iron (III) oxide deposited for the 0.8-mm-wide and the 4.0-mm-wide channels. Similar values of $q_e$ were calculated for all the channels regardless of width due to the concentration of arsenite in the solution being lower than the value required for a complete reaction between arsenite and iron (III) oxide. Note that although channel widths narrower than 0.8 mm can be manufactured, if the channel width is too narrow, the pre-treatments, including the hydrolysis and iron (III)-oxide binding, which require several repeated elution processes, will still take a much longer time. Thus, we chose an optimized channel width of ~ 0.8 mm for efficient pre-treatment processes and then used a rate controller to maximize the filtration power (S2 Fig).

## Isotherm study

The maximum removal capacity of a manufactured filter will vary with its channel width. In order to determine the maximum capacities of our manufactured filters, we prepared contaminated water samples with initial As(III) concentrations ranging from 10 mg/L to 130 mg/L by soaking the filtering columns in the As(III) solution for 3 hours, after which we investigated the effect of channel width on the removal arsenic. Note that this range of As(III) concentrations is far higher than WHO guideline concentration of 0.01 mg/L. As the initial concentration of As(III) was increased, the removal percentage of each filter was monitored (Fig 6A). While ca. 95% removal rate was maintained for the filters with narrower channels (0.8 mm and 1.0 mm), the removal rates for the filters with channel widths of 1.8 mm and 4.0 mm quickly became saturated and rapidly decreased. This result confirms that the maximum adsorption capacity of the filter increases as the surface area and the amount of iron-oxide loading in the filter are increased.

To determine the adsorption capacity of As(III) in filters with different channel widths, we conducted an equilibrium study, for which we adopted the Langmuir model. The pH of the solution was adjusted to 8.0, and the experiment was conducted at room temperature for 3-hour contact times. The model assumes that the adsorption is monolayer-wise onto a homogeneous surface with a finite number of identical sites. The mathematical expression

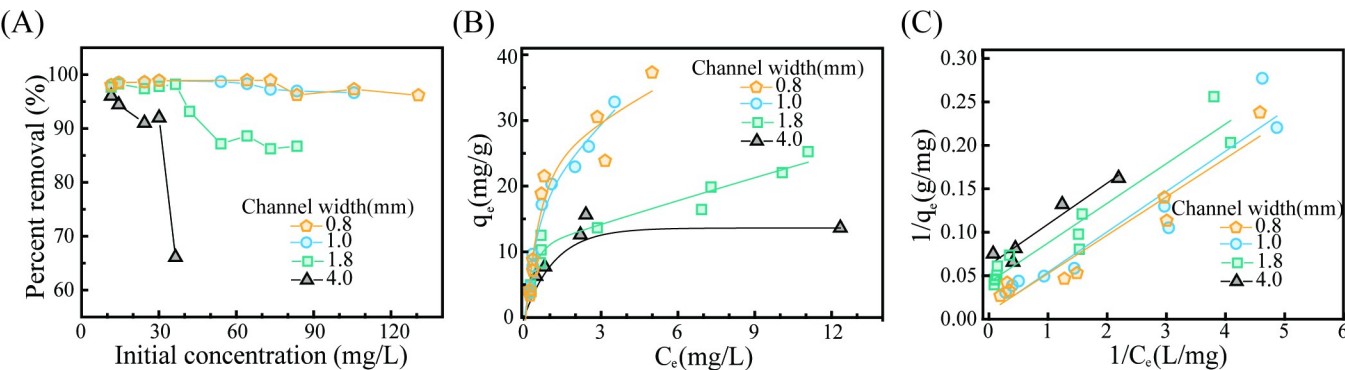

**Fig 6. Equilibrium studies for As (III) removal.** (A) As (III) removal percentage as a function of the initial concentration of the model contaminated solution. (B, C) Isotherm plots at different initial As (III) concentrations as predicted by using the Langmuir model.

describing this model is as follows:

$$q_e = q_{max}\frac{bC_e}{1 + bC_e},\tag{5}$$

where $q_e$ is the amount of As(III) adsorbed per unit weight of iron (III) oxide at equilibrium, $C_e$ is the equilibrium concentration of As(III) in the solution, $q_{max}$ is the maximum amount of As(III) per unit weight of iron (III) oxide (mg/g), and b is the Langmuir constant related to the affinity of the binding sites (L/mg). The relationship between $C_e$ and $q_e$ for filters with different channel widths is described in Fig 6B. For wider channels, $q_e$ stopped increasing at low $C_e$. However, for narrower channels, $q_e$ reached equilibrium at a relatively higher $C_e$. The values of $q_{max}$ and b were determined from a linear plot of $1/q_e$ versus $1/C_e$ (Fig 6C). The value of $q_{max}$ for filters with 0.8-mm-wide channels was 125.0 mg/g, 7 times higher than that for filters with 4.0-mm-wide channels (S2 Table).

Despite the differences in channel width, the filtration mechanism for all filters was a chemical reaction between the iron (III) oxide and arsenic. Nevertheless, significant differences in the values of $q_{max}$ were observed for filters with different channel widths. Such differences were assumed to have been due to the different amounts of iron (III) oxide deposited during the loading process. The amount of iron-hydroxide solution used for the loading process was the same for all filters regardless of the surface area inside. Therefore, the iron-oxide layers formed in filters with larger surface areas should be thinner than the iron-oxide layers formed in filters with smaller surface areas. The value of $q_{max}$ was observed to increase as the channel width was increased to 1.0 mm, and the Langmuir constant was observed to decrease as the channel became narrower.

At this point, we need to compare our results with the results for previously reported filters. Indeed, various adsorbents have been used to remove As (III) [47–51]. As can be seen in Table 1, the iron (III) oxide used as an absorbent in the 3D, printed PLA filter in this study showed superior performance with a higher maximum adsorption capacity of As (III) than the adsorbents used in the previously reported studies. Even when iron (III) oxide was used in such studies, various maximum adsorption capacities were reported depending on which supporting material was used. As shown in Table 2, various maximum adsorption capacities [16, 17, 19, 52, 53] that depended on the supporting material have been reported for an iron-oxide absorbent, and the 3D, printed PLA filter used in this study was found to be very good for removing As (III) from contaminated water.

In this 3D, printed filtration system, hydrostatic pressure pushing the contaminated water through the filter is the only driving force. Because the sizes of the meshes alter the resistance to eluent flow through the columns, the flow rates were varied. As shown in Fig 7A, narrower channels naturally slowed the flow through the filter by a factor of up to 10 times, i.e., from

**Table 1. Adsorption capacities of As (III) reported for difference adsorbents.**

| Adsorbent | Concentration range (mg/L) | pH | Adsorption capacity (mg/g) | References |
|---|---|---|---|---|
| hyperbranched polyethylenimine (HPEI)-modified cellulose fiber | 20–250 | 7 | 54.13 | [47] |
| iron-chitosan composite | 1–10 | 7 | 16.15 | [48] |
| copper (II)-oxide nanoparticles | 0.5–1 | 8 | 1.086 | [49] |
| magnetic $Fe_3O_4$/sugarcane bagasse-activated carbon composite | 2–120 | 8 | 6.69 | [50] |
| thiol functionalized sugarcane bagasse | 0.1–20 | 7 | 28.57 | [51] |
| 3D-printed PLA- coated Iron (III) oxide | 0.4–20 | 8 | 129.87 | 0.8-mm channel widths, this study |

**Table 2. Adsorption capacities of As (III) with iron oxide for different supporting materials.**

| Media support | Concentration range (mg/L) | pH | Adsorption capacity (mg/g) | References |
|---|---|---|---|---|
| glass fiber | 0.144–17.5 | 7 | 11 | [52] |
| carbon nanotubes | 0.1 | 4 | 1.723 | [16] |
| activated carbon | 20–22 | 6 | 51.3 | [17] |
| geopolymers | 0–0.12 | 7.6 | 0.95 | [53] |
| activated alumina | 1.6–2.3 | 12 | 0.378 | [19] |
| 3D-printed PLA | 0.4–20 | 8 | 129.87 | This study |

56.2 mL/s to 6.6 mL/s in the linear flow rates for channel widths of 4.0 mm and 0.8 mm, respectively. With these given flow rates, the elusion time for 500 mL will be quick, i.e., 9 s and 76 s for the filters with channel widths of 4.0 mm and 0.8 mm, respectively. For adsorptive filtration systems, however, sufficient elusion time for the interactions between arsenic-contaminated water and the iron-oxide particles to take place is required. When the model contaminated water of 500 mL with a concentration of 20 mg/L of arsenic was instantly filtered, the removal rate, even for the filter with a channel width of 0.8 mm, remained < 48% (Fig 7B), in contrast to the result in Fig 6A. Noted that the columns in Fig 6A were soaked and allowed to remain motionless for 3 hours to yield the maximum adsorption while the experiments, whose results are shown in Fig 7, were performed under the same condition as the filtration process; the contaminated water flowed through the filters hydrostatically. Considering that the concentration of arsenic in contaminated groundwater, in practice, is ~0.01 mg/L, a value of 48% for the water contaminated with arsenic at a concentration of 20 mg/L is still very effective for an As filtration system. Such a system can purify a volume of water equivalent to about 950 L without replacing the filter when water contaminated with 0.01 mg/L arsenic (corresponding to the WHO guideline as a drinkable water) is targeted. In other words, one adult drinks about 2 L per day, so such a system would provide an amount of water that can be used continuously for about 500 days.

When the target water is extremely contaminated, a rate controller can be adopted to maximize the filtration power (S2 Fig). A rate controller was designed to have an internal screw thread connected tightly to the filter (S2 Fig), and by varying the mesh design in the outlet of the rate controller, the retention time could be controlled. As shown in Fig 7A (blue dots-line),

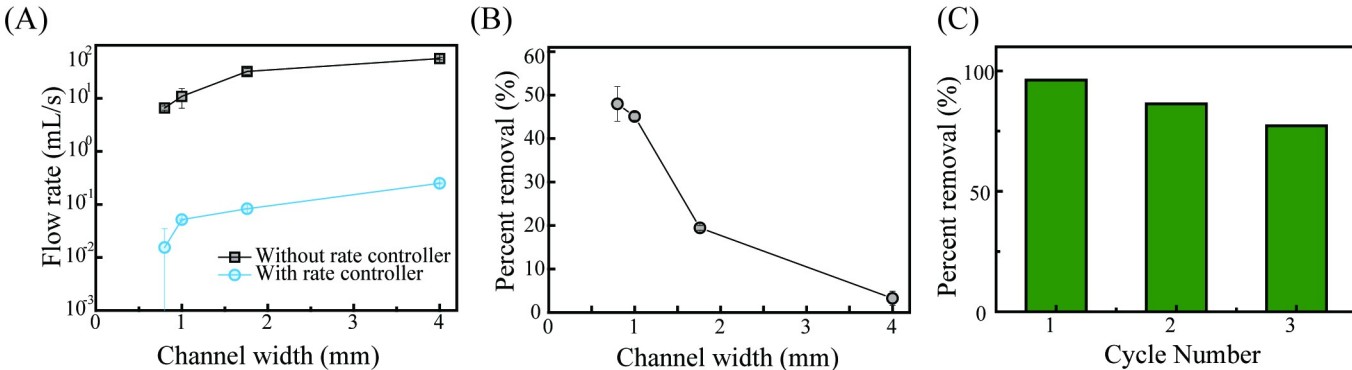

**Fig 7. Filtration tests for As (III) removal: flow rates, removal rates and recycling test.** (A) Flow rates with and without a rate controller as functions of the channel width. (B) Removal rates with the model contaminated water flowing through our 3D, printed adsorptive filters. In the experiment, 500 mL of water with an arsenic concentration of 20 mg/L was used. (C) Removal rates of recycled filters. A filter with a channel width of 0.8 mm was used, and a total of 15 mL of solution with an arsenic concentration of 20 mg/L was used. The filter with a channel width of 0.8 mm was tested using 15 mL of model contaminated water with an As (III) initial concentration of 20 mg/L.

the flow rate through a filter with a rate controller was much slower than that through a filter without a rate controller; reaction times under the given conditions were increased by a factor of 1000. Armed with this rate controller, the inverse relationship between the time (the flow rate) and the efficiency (the removal rate) of this filtration system can be balanced.

This study was aimed at producing cost-effective water purification filters for arsenic removal that can be used in resource-limited regions, so the economic aspect of such filters is important. In this context, we further tested whether this filter could be recycled. As described in Experimental, the adsorbent (iron (III) oxide)-Ar (III) complex layer from the filter, which had a removal rate of 96.2% for the first adsorption from a total of 15 mL of solution with an As (III) initial concentration of 20 mg/L, was treated using a strong acid solution; then, iron (III) oxide was re-deposited from a freshly prepared solution. As shown and confirmed in Fig 7C, the measured removal rates of As (III) were more than 86% and 77%, which represent sufficient adsorption capacities, even after the 2$^{nd}$ and the 3$^{rd}$ repeated recycling processes, respectively. For the 3D printer with a 0.4-mm nozzle and a 0.2-mm per layer height used in this study, 106 minutes were required to build a single filter with a channel width of 0.8 mm. The cost of chemicals and PLA filaments used in the fabrication for a single filter was estimated to be about USD 0.5 dollar. Although the time required to fabricate a single filter is not so fast, once the adsorptive filter has been made, it can be used and re-used without additional devices (i.e., pipes or motors). In sum, it is more affordable and more accessible, making it more suitable for use in resource-limited regions.

## Conclusion

In this report, we proposed the first 3D, printed water filter for the removal of As(III). Our proposed filter has a simple design and can easily be produced at low cost. Simple changes in the architecture of the filter had the following effects: As the channel width became narrower, corresponding to an increase in the internal surface area of the 3D printed filter, (1) more iron (III) oxide (adsorbent) was deposited on the inner surface of the filter, and (2) the flow rate under atmospheric conditions (without external pumping) decreased. The larger deposition of iron (III) oxide and the decreased flow rate led to better removal of arsenic. The effect of the internal surface area on arsenic removal was investigated thermodynamically and kinetically in a batch study. Filters with larger internal surface areas showed better purification performances compared to ones with smaller surface areas. A column study was performed to investigate the effect of the internal surface area on adsorptive filtration and showed better arsenic removal in adsorptive filtration. The study also showed that a filter with a larger internal surface area was more effective in adsorptive filtration than one with a smaller surface area. Because the flow rate of the water through the filter can be controlled by the architecture of the filter itself, extra pumping is not necessary. The compact filter presented in this report should serve well as a small-scale water-purification unit that can be easily carried to any site having water contaminated with arsenic.

## Supporting information

**S1 Fig.** Binding energy curves of the elements on the filter's surface: (A) O (1s) spectra, (B) C (1s) spectra, (C) Fe (2p) spectra, (D) As (3d) spectra.
(TIF)

**S2 Fig. 3D printed PLA filter equipped with a rate controller.**
(TIF)

**S1 Table. Kinetic parameters for As (III) removal by filters with four different channel widths.**
(DOCX)

**S2 Table. Isotherm parameters for As (III) removal by filters with four different channel widths.**
(DOCX)

**S3 Table. Reusability of the 3D-printed filter.**
(DOCX)

**S4 Table. Binding energies, percentage of total peak area, and assignment of atoms of XPS spectra of O (1s), C (1s), Fe (2p) and As (3d) from PLA, iron (III) oxide modified PLA filter, and iron (III) modified PLA filter after filtration.**
(DOCX)

**S1 File. Video for the Iron hydroxide loading process into a filter.**
(MP4)

## Author Contributions

**Conceptualization:** Kihoon Kim, Monica Cahyaning Ratri, Kwanwoo Shin.

**Investigation:** Kihoon Kim, Giho Choe, Myeongyun Nam, Daehyoung Cho.

**Writing – original draft:** Kihoon Kim.

**Writing – review & editing:** Monica Cahyaning Ratri, Kwanwoo Shin.

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
