## [Decision Letter · Decision Letter 0]

3 Jan 2020

PONE-D-19-28310

Three-dimensional printed, water-filtration system for economical and on-site arsenic removal

PLOS ONE

Dear Dr Shin,

Thank you for submitting your manuscript to PLOS ONE. After careful consideration, we feel that it has merit but does not fully meet PLOS ONE’s publication criteria as it currently stands. Therefore, we invite you to submit a revised version of the manuscript that addresses the points raised during the review process.

We would appreciate receiving your revised manuscript by Feb 17 2020 11:59PM. To enhance the reproducibility of your results, we recommend that if applicable you deposit your laboratory protocols in protocols.io, where a protocol can be assigned its own identifier (DOI) such that it can be cited independently in the future. For instructions see: http://journals.plos.org/plosone/s/submission-guidelines#loc-laboratory-protocols

We look forward to receiving your revised manuscript.

Kind regards,

Zhi Zhou, Ph.D.

Academic Editor

PLOS ONE

Journal Requirements:

Reviewers' comments:

Reviewer's Responses to Questions

**Comments to the Author**

1. Is the manuscript technically sound, and do the data support the conclusions?

Reviewer #1: Yes

Reviewer #2: Partly

2. Has the statistical analysis been performed appropriately and rigorously? 

Reviewer #1: Yes

Reviewer #2: Yes

3. Have the authors made all data underlying the findings in their manuscript fully available?

Reviewer #1: Yes

Reviewer #2: No

4. Is the manuscript presented in an intelligible fashion and written in standard English?

Reviewer #1: Yes

Reviewer #2: No

5. Review Comments to the Author

Reviewer #1: A 3D printed, water-filtration system was developed for arsenic removal, and the internal surface areas of filters were modified with iron (III) oxide to improve adsorption capacities. The research was interesting and deserved to be accepted after some revisions:

1. In Introduction section, the assembly of novel 3D architectures as novel adsorbents should be comapred, and some recently published related references should be cited and discussed: Mater Chem Phys, 2019, 231, 105-108; Microchemical Journal, 2020, 152, 104288; Chem Eng J, 2019, 363, 107-119; etc.

2. Why strong acid was used for thr hydrolysis of PLA to yield hydroxyl and carboxylic groups? Strong base would be more efficient;

3. Why Fe(III) was introduced into the filter? If ion exchange dominated the adsorption, sodium salt based modification would be better;

4. The adsorption mechanism should be verified by more detailed characterizations including XRD, XPS for the adsorbents before and after adsorption.

5. A table of comparison of the adsorbent with previously reported results should be provided.

Reviewer #2: This manuscript reports a 3D printed water filtration system for Arsenic removal. 3D printed filters were used as the support for adsorbents iron-oxide. The work is interesting and important. However, it should be significantly improved before publications by addressing the following concerns.

1. The paper highlights “economical”, “low-cost” in the title and abstract. However, there is no any cost analysis. Complete cost analysis should be done.

2. “As the channel gets narrower, removal of arsenic is faster and better”. Why did not the authors further reduce the channel width? The interfacial surface areas and volumes of the channels with different channel widths were reported. However, these parameters should also be compared with other supports/nanostructures.

3. What were the channel lengths/heights? Did they affect the adsorption performance?

4. The adsorption performance of the filter in this study should be compared with the supports developed by other researchers.

5. The control study without any support should be done for comparison.

6. What are stability and reusability of the filtration system? Such work should be added.

7. What is the distribution of the adsorbents across the filter/channel? Such data should be provided.

8. There are many grammatical errors in the manuscript.

6. PLOS authors have the option to publish the peer review history of their article (what does this mean?). If published, this will include your full peer review and any attached files.

Reviewer #1: No

Reviewer #2: No

---

## [Author Response · Author response to Decision Letter 0]

1 Mar 2020

Response to reviewer comments:3

Reviewer #1 : 

A 3D printed, water-filtration system was developed for arsenic removal, and the internal surface areas of filters were modified with iron (III) oxide to improve adsorption capacities. The research was interesting and deserved to be accepted after some revisions:

Reply:

We are delighted that Reviewer #1 is so positive, and agree with the Reviewer that this topic engaged by the paper is interesting. At a same time, we have taken the Reviewer’s comments seriously, and we have sought to correct the gap left by this oversight. We thank the Reviewer for the suggestions made, which have helped us to add a number of important additional data and details. 

In Introduction section, the assembly of novel 3D architectures as novel adsorbents should be compared, and some recently published related references should be cited and discussed: Mater Chem Phys, 2019, 231, 105-108; Microchemical Journal, 2020, 152, 104288; Chem Eng J, 2019, 363, 107-119, etc.

Reply:

 We appreciate the careful consideration of the reviewer with respect to the related recent literatures. We have incorporated discussion of the three papers into the text (References 10, 11, and 43), noting that, in particular, the reusability and the improved performance in adsorptive filtration systems. 

Why strong acid was used for the hydrolysis of PLA to yield hydroxyl and carboxylic groups? Strong base would be more efficient;

Reply:

 We appreciate the Reviewer for raising this issue, and have included further information regarding the reason of the hydrolysis condition in the manuscript. While the final purpose of the hydrolysis here is not only for the chain scission, but also for the generation of binding sites, hydroxyl and carboxylic acid, which are necessary to deposit Fe(OH)3 particles. The hydrolysis of PLA will be surely occurred in basic condition, as the Reviewer pointed out, but carboxylate groups will be formed in the basic condition instead.

Why Fe(III) was introduced into the filter? If ion exchange dominated the adsorption, sodium salt based modification would be better;

Reply:

We thank the Referee for bringing this ambiguity to our attention. This is a point that we should clarify and emphasize. As we explained in the text, iron oxide is one of the most promising adsorbents with its high affinity to arsenite. More importantly, these iron oxide minerals are basically iron rust powder, which is abundant in any household. Since this research was to provide an alternative way to reduce the arsenic level in water in resource-limited areas, we believe that the iron oxide would be a highly effective, yet economic way to remove the harmful arsenite, when it combined with the customized 3D fabrication. This discussion has been added to the manuscript. 

The adsorption mechanism should be verified by more detailed characterizations including XRD, XPS for the adsorbents before and after adsorption.

Reply:

We indisputably agree that the suggested data will be of great help to strengthen the manuscript by clarifying the adsorption mechanism. Indeed, the XPS spectra do correlate with every step of filtration process, 1) the acid induced oxidation (S1 Fig A), 2) the carboxylic acid formation upon the oxidation, 3) the binding of iron oxide particles, and 4) the adsorption of As (III) on the iron oxide particles. The manuscript has been revised to include this information, and have added representative spectra to the Supporting Information (S1 Fig and S4 Table), as suggested. 

A table of comparison of the adsorbent with previously reported results should be provided.

Reply:

 We appreciate the Reviewer's suggestion, which has given us an opportunity to strengthen the manuscript. We have thoroughly searched a number of literatures, regarding on the adsorption capacities of As (III) in different adsorbents and media supports and have listed in Tables 1 and 2. The range of references drawn on in this result has been extended with the inclusion of a number of papers, reference 8, 16-17, 19 and 47-53 being among the many added and compared. 

We thank the Reviewer for his or her careful reading of our manuscript and for the suggestions made, which have helped us to add a number of clarifications and important publications to those discussed. 

Reviewer #2;

This manuscript reports a 3D printed water filtration system for Arsenic removal. 3D printed filters were used as the support for adsorbents iron-oxide. The work is interesting and important. However, it should be significantly improved before publications by addressing the following concerns.

Reply:

We are delighted that Reviewer #2 is so positive, and agree with the Reviewer that this topic engaged by the paper is interesting and important. We thank the Reviewer for his or her attentive review of our paper, and for the suggested changes which we feel have broadened the scientific base drawn on by this manuscript. 

The paper highlights “economical”, “low-cost” in the title and abstract. However, there is no any cost analysis. Complete cost analysis should be done.

Reply:

 We appreciate this comment, as it highlights an important aspect of how our 3D printed filtration system is cost-effective in this context. Indeed, the cost analysis was added in text. Note that the estimated price was given by the sum of the prices of the 3D printable PLA filament, the hydroxide solution and the iron (III) chloride used to fabricate a single filtration system. 

“As the channel gets narrower, removal of arsenic is faster and better”. Why did not the authors further reduce the channel width? 

Reply:

 While the channel width is a key parameter, determining the filter’s capacity, as described and detailed in the text and S1 Table, another important factor to be optimized is the flow rate of filtration process. In this 3D printed filtration system, hydrostatic pressure is the only driving force, causing the contaminated water to flow through the filter. Therefore, depending on the mesh sizes, the flow rates are altered. As described in Figs 6A and B, a slower flow of water increases the adsorption of arsenic was increased. However, it is also true that if the channel is too narrow, then the pre-treatments, including the hydrolysis and iron (III) oxide binding which are also requiring several repeated elution processes, will take much longer time. Instead, we choose an optimized channel width of 0.8 mm for efficient pre-treatment processes, and then additionally adopt a rate controller for the maximized filtration power. We have included additional discussion in the manuscript to clarify this important issue. 

The interfacial surface areas and volumes of the channels with different channel widths were reported. However, these parameters should also be compared with other supports/nanostructures.

Reply:

 We appreciate the Reviewer #2 for raising this issue. In line with the similar question, raised by the Reviewer #1, we have included further information regarding other adsorbents reported. We have thoroughly searched a number of literatures, regarding on the adsorption capacities of As (III) in different adsorbents and media supports and have listed in Tables 1 and 2. The range of references drawn on in this result has been extended with the inclusion of a number of papers, reference 8, 16-17, 19 and 47-53 being among the many added and compared. 

What were the channel lengths/heights? Did they affect the adsorption performance?

Reply:

 We thank the Reviewer for pointing this out. We have provided the column dimension (a 40 mm long height and a 30-mm outer diameter) in Fig 1A. We have added the information in the text as well. The change in channel height is related to the internal surface area and total channel volume of the filter, therefore, doubling the height of column, for instance, will double the surface area, surely enhancing the adsorption performance per a single filtration process. 

The adsorption performance of the filter in this study should be compared with the supports developed by other researchers.

Reply:

 As also answered in the response to the Referee #1, we have included the comparisons in Tables 1 and 2, highlighting that our system, indeed, outperforms any other systems developed previously. 

The control study without any support should be done for comparison.

Reply:

 While it is not clear whether the support the Reviewer mentioned indicates PLA column or iron (III) oxides, we infer that the comparison should be done with and without iron (III) oxides in this context. In agreement with the Reviewer who suggested the additional control experiments, we have performed XPS experiments (as shown in S1 Fig). We found the clear peak from the adsorbed arsenic from the iron (III) oxide modified filter, while no hint of arsenic ion was found in the specimen from non-treated PLA filters.

What are stability and reusability of the filtration system? Such work should be added.

Reply:

 We thank the Reviewer for bringing this to our attention. Up on the request from the Reviewer, we have developed the recycling protocol; after the initial filtration of arsenites, the used filter was then immersed in 20 % H2SO4 solution to detach the ion-complexes from the filter surface. The repeated filtrations was then performed after the repeated process of iron (III) oxide coating. Even after 3 times of recycling, the arsenic removal rates from a total of 15 mL of solution with an As (III) concentration of 20 mg/L on were slightly lowered but maintained at well above 77 %, in comparison to the initial removal rate of 96.2 %. In real life, the arsenic contents in drinking water (even for the contaminated water) should be much lower (<0.01 mg/L), so we believe that this filtration system can be applicable for many times. As mentioned in text (p. 17), such system can purify a volume of water equivalent to about 950 L without replacing the filter when water contaminated with 0.01 mg/L arsenic (corresponding to the WHO guideline as a drinkable water) is targeted. In other words, one adult drinks about 2 L per day, so such a system would provide an amount of water that can be used continuously for about 500 days. In this context, Figure (Fig 6C) and discussion have been added in the revised manuscript. 

What is the distribution of the adsorbents across the filter/channel? Such data should be provided. 

Reply:

 We appreciate this comment, and have revised the manuscript to improve the discussion regarding the distribution of the adsorbents (iron (III) oxides) on PLA surface. As visualized in Fig 3A, the adsorbents were evenly distributed on the oxidized PLA surface. 

There are many grammatical errors in the manuscript.

Reply:

 Our revised manuscript has been proofread by a professional editor, and corrected a number of grammatical errors. We believe that our final manuscript has been much improved.

We thank the Reviewer for his or her attentive review of our manuscript, and for the suggested changes which we feel have allowed us to improve and clarify the work.

---

## [Decision Letter · Decision Letter 1]

25 Mar 2020

Three-dimensional, printed water-filtration system for economical, on-site arsenic removal

PONE-D-19-28310R1

Dear Dr. Shin,

We are pleased to inform you that your manuscript has been judged scientifically suitable for publication and will be formally accepted for publication once it complies with all outstanding technical requirements.

With kind regards,

Zhi Zhou, Ph.D.

Academic Editor

PLOS ONE

Additional Editor Comments (optional):

Reviewers' comments:

Reviewer's Responses to Questions

**Comments to the Author**

1. If the authors have adequately addressed your comments raised in a previous round of review and you feel that this manuscript is now acceptable for publication, you may indicate that here to bypass the “Comments to the Author” section, enter your conflict of interest statement in the “Confidential to Editor” section, and submit your "Accept" recommendation.

Reviewer #1: All comments have been addressed

2. Is the manuscript technically sound, and do the data support the conclusions?

Reviewer #1: Yes

3. Has the statistical analysis been performed appropriately and rigorously? 

Reviewer #1: Yes

4. Have the authors made all data underlying the findings in their manuscript fully available?

Reviewer #1: Yes

5. Is the manuscript presented in an intelligible fashion and written in standard English?

Reviewer #1: Yes

6. Review Comments to the Author

Reviewer #1: The manuscript was revised as necessary, and it was wel revised. The manuscript could be accepted as it is.

7. PLOS authors have the option to publish the peer review history of their article (what does this mean?). If published, this will include your full peer review and any attached files.

Reviewer #1: No

---

## [Editor Report · Acceptance letter]

1 Apr 2020

PONE-D-19-28310R1 

Three-dimensional, printed water-filtration system for economical, on-site arsenic removal 

Dear Dr. Shin:

I am pleased to inform you that your manuscript has been deemed suitable for publication in PLOS ONE. Congratulations! Your manuscript is now with our production department. 

With kind regards,

on behalf of

Dr. Zhi Zhou 

Academic Editor

PLOS ONE